# Transcriptomic analysis of Pacific white shrimp (*Litopenaeus vannamei*, Boone 1931) in response to acute hepatopancreatic necrosis disease caused by *Vibrio parahaemolyticus*

**Adrián E. Velázquez-Lizárraga**[1], **José Luis Juárez-Morales**[2], **Ilie S. Racotta**[3], **Humberto Villarreal-Colmenares**[4], **Oswaldo Valdes-Lopez**[5], **Antonio Luna-González**[6], **Carmen Rodríguez-Jaramillo**[7], **Norma Estrada**[2]*, **Felipe Ascencio**[1]*

**1** Laboratorio de Patogénesis Microbiana, Centro de Investigaciones Biológicas del Noroeste, S. C. (CIBNOR), La Paz, Baja California Sur, México, **2** Programa de Cátedras CONACyT, Centro de Investigaciones Biológicas del Noroeste, S. C. (CIBNOR), La Paz, Baja California Sur, México, **3** Laboratorio de Metabolismo Energético, Centro de Investigaciones Biológicas del Noroeste, S. C. (CIBNOR), La Paz, Baja California Sur, México, **4** Parque de Innovación Tecnológica, Centro de Investigaciones Biológicas del Noroeste, S. C. (CIBNOR), La Paz, Baja California Sur, México, **5** Departamento de Bioquímica, Facultad de Estudios Superiores – Universidad Autónoma de México, Tlalnepantla de Baz, Estado de México, México, **6** Departamento de Acuacultura. Instituto Politécnico Nacional-Centro Interdisciplinario de Investigación para el Desarrollo Integral Regional, Unidad Sinaloa (IPN-CIIDIR Sinaloa), Guasave, Sinaloa, México, **7** Laboratorio de Histología, Centro de Investigaciones Biológicas del Noroeste, S. C. (CIBNOR), La Paz, Baja California Sur, México

\* ascencio@cibnor.mx (FA); nestrada@cibnor.mx (NE)

## Abstract

Acute hepatopancreatic necrosis disease (AHPND), caused by marine bacteria *Vibrio Parahaemolyticus*, is a huge problem in shrimp farms. The *V. parahaemolyticus* infecting material is contained in a plasmid which encodes for the lethal toxins PirAB$^{Vp}$, whose primary target tissue is the hepatopancreas, causing sloughing of epithelial cells, necrosis, and massive hemocyte infiltration. To get a better understanding of the hepatopancreas response during AHPND, juvenile shrimp *Litopenaeus vannamei* were infected by immersion with *V. parahaemolyticus*. We performed transcriptomic mRNA sequencing of infected shrimp hepatopancreas, at 24 hours post-infection, to identify novel differentially expressed genes a total of 174,098 transcripts were examined of which 915 transcripts were found differentially expressed after comparative transcriptomic analysis: 442 up-regulated and 473 down-regulated transcripts. Gene Ontology term enrichment analysis for up-regulated transcripts includes metabolic process, regulation of programmed cell death, carbohydrate metabolic process, and biological adhesion, whereas for down-regulated transcripts include, microtubule-based process, cell activation, and chitin metabolic process. The analysis of protein-protein network between up and down-regulated genes indicates that the first gene interactions are connected to oxidation-processes and sarcomere organization. Additionally, protein-protein networks analysis identified 20-top highly connected hub nodes. Based on their immunological or metabolic function, ten candidate transcripts were selected to measure their mRNA relative expression levels in AHPND infected shrimp hepatopancreas by RT-qPCR. Our results indicate a close connection between the immune and metabolism

**Data Availability Statement:** All relevant data are within the paper and its Supporting Information files.

**Funding:** This research was supported by grants of Consejo Nacional de Ciencia y Tecnología (CONACyT) No. 247567 to NE and from CIBNOR (PAC) To FA. AEVL have a scholarship from CONACyT No. 466639. The funders had no role in study design, data collection, and analysis, decision to publish, or preparation of the manuscript.

**Competing interests:** The authors have declared that no competing interests exist.

systems during AHPND infection. Our RNA-Seq and RT-qPCR data provide the possible immunological and physiological scenario as well as the molecular pathways that take place in the shrimp hepatopancreas in response to an infectious disease.

## Introduction

Aquaculture of *Litopenaeus vannamei* (whiteleg shrimp) along with *Penaeus monodon*, is one of the most important cultivated shrimp species with a significant economic value that generates billions of USD every year. Shrimp farming has been a growing industry since the '80s whose worldwide production rose to 4.5 million tons in 2016 [1]. Despite the success of shrimp production worldwide, this industry has had several setbacks due to viral and bacterial diseases which have caused until 2014, 118 million USD in losses [2,3].

One of the primary bacterial infections affecting shrimp farming is the acute hepatopancreatic necrosis disease (AHPND) formerly called "early mortality syndrome" (EMS), the first reported AHPND outbreak occurred in China in 2009 which later spread out through to Malaysia, Thailand, Vietnam, and Mexico [4].

AHPND causes 90–100% mortality in tainted shrimp ponds; the clinical signs of infected shrimp include lethargy, erratic swimming, empty stomach and intestine, and un-colored hepatopancreas. The first and foremost shrimp organ infected by AHPND is the hepatopancreas, a crucial digestive gland involved in controlling systemic metabolism, digestion, storage of nutrients, and whose multiple physiological and immunological functions are affected during infection. The etiological agent causing AHPND are bacteria of the genus *Vibrio*, being *V. parahaemolyticus* the primary infection vector. This bacteria species contains a plasmid that encodes for the lethal toxins PirAB$^{Vp}$ that cause sloughing of hepatopancreas epithelial tubule cells (F, R, and B cells) in early infection stages, followed by necrosis, atrophy, and massive hemocytes infiltration [5–8].

Although several molecular tools and histopathological tests have been developed and are currently available to detect AHPND [9–11], the physiological and biochemical alterations caused by AHPND in the organism are far from understood.

Previous studies that examined the stomach response of *P. monodon* against AHPND revealed 141 up-regulated immune-related genes of which functional categories include: antimicrobial peptides, proteinase/proteinase inhibitors, signal transduction pathway, proPO system, and oxidative stress proteins, among others [12].

Moreover, the differentially expressed miRNAs of *L. vannamei* in non-infected vs. infected shrimp with *V. parahaemolyticus* suggests that miRNAs modulate several immune processes during AHPND [12], and the response of hemocytes challenged with *V. parahaemolyticus* shows that apoptosis is one of the major processes that take place during pathogenesis by affecting anti-apoptosis gene expression [13,14].

Lastly, using the oriental prawn *Exopalaemon carinicauda* as a host-pathogen model, Ge and collaborators showed that 127 immune response genes were up-regulated in the prawn hepatopancreas with AHPND, indicating that the first response of defense mechanisms that take place during infection [15].

Even though there has been a considerable amount of work and efforts put into understanding the biology and controlling the AHPND, to date, there has not been a published transcriptional profile of *L. vannamei* hepatopancreas with AHPND emulating the natural course of infection. This study aims to investigate the transcriptional mRNA response of the

*L. vannamei* hepatopancreas during AHPND infection, to gain a better understanding of AHPND shrimp pathogenesis and to find new genomic approaches to address this aquaculture problem.

## Materials and methods

### Ethics statement

All shrimp used for experimental work was handled in accordance with the Official Mexican Standard protocols (NOM-062-ZOO-1999). Post larval *Litopenaeus vannamei* specimens were obtained from Laboratorio de Larvas Granmar S.A. de C.V. and transported to aquarium tanks at CIBNOR with all permits issued by the federal agency CONAPESCA. Juvenile rearing was done following the hatchery's standard procedures until the shrimps weighed 12 g. *L. vannamei* is not considered as an endangered or protected species.

### Biological material

Healthy *L. vannamei* weighing about 12.26 ± 0.022 g were maintained in 1000 L PVC tanks filled with filtered (1 μm) and aerated seawater, salinity of 35 PSU, at 25 ˚C. Shrimp were acclimatized and fed ad libitum daily for one week, with commercial feed (Camaronina, Purina, Mexico City). *V. parahaemolyticus* IPNGS16 was isolated and characterized from dying shrimp obtained from farms in Guasave, Sinaloa, Mexico [16].

### *Vibrio parahaemolyticus* IPNGS16 characterization

The bacterial strain was reactivated in tryptic soy broth (BD Difco, Sparks, MD) with 2.5% NaCl (TSB+) at 28 ˚C for 16 h and 120 rpm. After incubation, 10 μL of the bacterial culture were inoculated in a petri dish with thiosulfate-citrate-bile salts-sucrose agar (TCBS) (BD Difco, Sparks, MD) by cross streak method and incubated at 30˚C for 24 h. A single colony was inoculated in TSB+ and cultured as mentioned above; a small aliquot was used for Gram staining to examine bacterial morphology. DNA extraction was performed as mentioned in Tyagi and collaborators [17]. Molecular identification of AHPND virulence genes *PHP* and *PirB* was done by end point PCR. PCR amplifications were performed in 25 microliter reactions: 2.5 mM MgCl$_2$, 0.2 mM dNTPs (each), 1X colorless buffer GoTaq Flexi DNA polymerase system (Promega, Madison, WI), 1.25 U GoTaq Flexi polymerase, 0.8 μM of each primer, 2 μL of DNA (50 ng). PCR primer sequences for the *PHP* gene are `forward (5'—TTCTCA CGATTGGACTGTCG—3')` and `reverse (5'—CACGACTAGCGCCATTGTTA—3')` [18]. For the *PirB* gene, we design the primer sequences: `forward (5'—TACGCCAAATGAGCC AGA—3')` and `reverse (5'—ACCAACTACGAGCACCCATC—3')`.

PCR cycling conditions were as follows: initial denaturalization 94 ˚C 2 min, 35 cycles of 95 ˚C 30 s, 60 ˚C 25 s and 72 ˚C 30 s, and a final extension of 72 ˚C 5 min. PCR products were separated in a 1.2% agarose gel (1X TAE) and stained with Red Gel 1X (Biotium, Fremont, CA). PCR products were sequenced in both directions by Sanger method [19]. The molecular weight of the *PirB* and *PHP* region amplified is 501 bp and 272 bp (S1 Fig) of plasmid pVPA3-1 (KM067908.1).

### Median lethal concentration (LC$_{50}$)

Shrimps were transferred to 40 L plastic tanks containing 20 L of filtered (1 μm) and aerated seawater, and salinity of 35 PSU. Ten organisms were placed in each tank and acclimatized for three days at 28 ˚C.

To calculate the median lethal concentration ($LC_{50}$), five bacterial cell concentrations were prepared; each concentration was tested in triplicate.

Briefly, *V. parahaemolyticus* stored in 30% glycerol at -80 ˚C, was reactivated in TSB+ medium and incubated for 16 h at 28 ˚C. Bacterial culture was then centrifuged at $2,600 \times g$ for 20 min at 4 ˚C; bacteria pellets were washed and resuspended in saline solution (2.5% NaCl). The bacterial solution was adjusted spectrophotometrically to an optical density of 1 at 600 nm (Bio-Rad, Smart-Spec 3000) [16].

Different concentrations of bacteria suspensions (350, 750, 1000, 1500, and 3000 CFU/mL) were applied to the shrimp culture tanks (20 L total volume) and incubated for 24 h. Shrimp mortality was recorded every 6 h by visual inspection in each tank, dead shrimps were removed from the tanks. Shrimp was grown in optimal conditions, however, no cleaning of the tanks was made during the challenging period and temperature was maintained at 28˚C to promote *V. parahaemolyticus* infection. $LC_{50}$ was determined at 24 hours using Probit analysis in IBM SPSS v23 (IBM, Armonk, NY)[20].

## Experimental design

Before bacterial challenge, experimental organisms were examined under the microscope to determinate molt stage based on the degree of setae development in uropods [21].

Organisms in the intermolt stage (C1) were placed in 20 L plastic tanks at 28 ˚C and 35 PSU under continuous aeration and left to acclimatize for three days. A total of 18 tanks (9 control and 9 experimental) were set up with ten organisms per tank. Experimental infection was carried out using a bacterial concentration of 660 CFU/mL ($LC_{50}$ at 24 h) per tank; experimental organisms were exposed to the bacteria for at least 72 h.

After bacterial exposure, hepatopancreas samples (n = 9) were collected at 0 hours (before infection), 3, 6, 12, 24, and 48 hours post-infection (hpi) for control and treated groups; around ~100 mg of each sample was placed in RNA-stabilizing buffer (70% $(NH_4)_2SO_4$, 25 mM sodium citrate, 10 mM EDTA, pH 5.2 adjusted with $H_2SO_4$) followed by progressive freezing for, 24 h at 4 ˚C, 24 h at -20 ˚C and permanent storage at -80 ˚C [22].

## Histopathology

Five organisms were fixed per time (0, 12, 24, 48 and 72 hpi), with 1 mL of Davidson's solution injected in the hepatopancreas [23]. After injection, whole mount shrimps were fixed with the same solution for 48 h. Subsequently, organisms were dehydrated in ethanol series of 70%, 80%, 90%, 100%, and xylene 100%. Afterward, tissues were embedded in paraffin (Paraplast X-TRA, Medline Industries Inc., Tolleson, AZ), longitudinally sectioned to 4 μm thickness and stained with hematoxylin-eosin. After staining, samples were examined in a BX41 Olympus microscope model tri-ocular BX41 (Olympus America Inc., Center Valley, PA), using a 40× and 100× mineral oil objectives. Images were captured using the digital image system Image-Pro Premier software v9.0 (Media Cybernetics Inc., Rockville, MD).

## Hepatopancreas RNA-seq

Total RNA from 24 h samples (control and infected groups) was extracted with Tri-reagent (Sigma-Aldrich, St. Louis, MO) according to manufacturer's protocol. RNA samples were quantified using Qubit 2.0 Fluorometer (Life Technologies, Carlsbad, CA), the RNA integrity was checked with 2100 Bioanalyzer (Agilent Technologies, Santa Clara, CA) only samples with an RNA integrity number (RIN) > 8.5 were used. Samples for each treatment (control and infected) were pooled into one tube each and sent for sequencing to GENEWIZ, LLC. (South Plainfield, NJ). mRNA was isolated from total RNA using Oligo d(T) beads and fragmented

for 15 min at 94 ˚C. First and second cDNA strands were synthesized and the 3'-ends were repaired and adenylated. Subsequently, sequencing adapters were ligated to cDNA fragments to prepare a cDNA library.

RNA sequencing library was prepared using the NEBNext Ultra RNA Library Prep Kit (New England Biolabs, Ipswich, MA) from Illumina following manufacturer's recommendations. The universal adapter was ligated to cDNA fragments, followed by index addition and library enrichment with limited PCR cycle. Sequencing libraries were validated using a DNA Chip on the Agilent 2100 Bioanalyzer (Agilent Technologies, Palo Alto, CA), and quantified using Qubit 2.0 Fluorometer (Invitrogen, Carlsbad, CA) as well as by quantitative PCR (Invitrogen, Carlsbad, CA) as well as by quantitative PCR (Applied Biosystems, Carlsbad, CA, USA).

The sequencing libraries were then multiplexed and clustered onto a flow cell; and loaded in the illumina HiSeq 2500 instrument (San Diego, CA) according to manufacturer's instructions. Samples were sequenced using a 2X100bp paired-end configuration. The HiSeq Control Software conducted the image analysis and base calling. Raw sequence data generated from Illumina HiSeq 2500 was converted into fastq files and de-multiplexed using illumina bcl2fastq v1.8.4.

## Transcriptome assembly and functional annotation

We used Trimmomatic v0.36 to trimmed and filtered the raw reads in order to remove adapters sequences, reads with unknown nucleotides and low-quality reads (Q ≤ 20) [24]. Clean reads were *de Novo* assembled with Trinity v2.4.0 using kmer size 31. Assembly quality was assessed with Trinity scripts. We aligned the trimmed reads vs. all transcripts assembled with Bowtie2 [25–27]. Transcripts assembled were annotated via Trinotate v3.0 pipeline (http://trinotate.github.io), with the following additional step, nucleotide transcripts sequences were examined with TransDecoder tool v5.0.2 (https://github.com/TransDecoder) to find the longest open reading frames. A UniProtKB database was built and filtered using the main invertebrate phyla (Arthropoda, Mollusca, Porifera, Cnidaria, Echinodermata, Platyhelminthes, Nematoda, and Annelida), the generated database was named UniProtKB[INV]. Nucleotide and protein sequences were aligned with BLASTp and BLASTx against SwissProt and UniProtKB[INV] databases. The SwissProt database was used to assign names of homolog protein and retrieve KEGG, GO and eggNOG annotations [28–32]. To identify protein domains HMMER v3.1b2 (http://hmmer.org) was used against Pfam 31.0 database [33,34]. Moreover, to predict signal peptides and transmembranal regions, we used software SignalP v4.1f and TMHMM v2.0c [35,36]. The previous results were built into an SQLite database.

## Transcript quantification and differential expression analysis

Transcript abundance was estimated using Trinity v2.4.0 downstream analysis pipeline [25]. Transcript abundance was estimated with Bowtie2 v2.3.4 and RSEM using both libraries (control and infected) [26,37]. The transcript expression matrices were built in a database, and the expression patterns were calculated using the Transcript per Kilobase Million (TPM) method [38,39]. The read counting was normalized to obtained relative expression. We used the False Discovery Rate (FDR) method to ensure the statistical significance of differentially expressed transcripts (DETs). Transcript clusters of our DETs of interest were generated using FDR≤0.01 and log2Ratio≥2 [40]. The resulting DETs clusters were annotated as mentioned above.

## GO term and KEGG pathway enrichment analysis of DETs

To select genes of interest from our DETs, a Gene Ontology (GO) enrichment analysis was conducted with 'GOseq' based on the hypergeometric test (P ≤ 0.01) value [41]. Additionally, a pathways enrichment analysis was performed using the KEGG Automatic Annotation Server (KAAS) [42].

## Network analysis and protein-protein interaction of DETs

Functional interactions of DETs were predicted using the GeneMANIA algorithm, a tool from the Cytoscape v3.6.0 software [43,44]. The parameters used were based on Gene Ontology (GO) under the term "biological process," and *Drosophila melanogaster* was used as a source species. Three different networks were constructed depending on their gene expression levels: up-regulated transcripts, down-regulated transcripts and a combination both. The networks were constructed based on different interaction levels: co-expression of genes, physical and genetic interactions, pathway participation, protein co-localization, protein domain similarity, and predicted protein interactions.

## Identification of hub genes and DETs clustering

The hub genes are represented as nodes with larger interactions into the network and were identified by calculating the node degree distribution [45,46]. The network that combined the up and down-regulated transcripts was used to perform a community analysis using cluster-Maker and the greedy algorithm [47–49]. The resulting clusters were analyzed by enrichment analysis using the DAVID tool [50].

## Gene selection and primer design

We selected ten candidate transcripts based on our differential expression analysis, these genes were selected because they are known to participate in the immune and metabolic response against AHPND. Primers for qPCR were designed using the primer3plus software, the list of primers are shown in Table 1. Primers at [25 nM] were synthesized at T4OLIGO (Irapuato, Guanajuato, Mexico) and purified by desalting method (DST) [51].

## Relative gene expression analysis

Total RNA extraction from hepatopancreas was conducted as mentioned above. RNA quantity and purity were assessed measuring the $A_{260nm}/A_{280nm}$ and $A_{260nm}/A_{230nm}$ ratio respectively using Nanodrop 1000 (Thermo Scientific, Chicago, IL). Additionally, RNA quality and 18S and 28S rRNA band integrity were verified by electrophoresis on a 1.2% Bleach gel [1X TAE] [52]. Afterward, 1 μg of total RNA was treated with 1 u of DNAse I amplification grade (Sigma Aldrich, St. Louis, MO). For first-strand synthesis we used ImPromp II reverse transcription system (Promega, Madison, WI) following manufacturer's instructions using 0.5 μg of oligo d (T) for mRNA capture, cDNA was stored at -80 ˚C until use.

Afterward, 1 μg of total RNA was treated with 1 μL of DNAse I (1 u/μL) amplification grade (Sigma Aldrich, St. Louis, MO). For first-strand synthesis we used ImPromp II reverse transcription system (Promega, Madison, WI) following manufacturer's instructions using 0.5 μg of oligo d(T) for mRNA capture, cDNA was stored at -80 ˚C until use.

Amplification efficiencies (E) for all primers used in this experiment were conducted by slope calculation of 4-fold serial dilutions starting with 150 ng/μL of cDNA on a threshold value of 0.034 and calculated automatically using the Rotor-Gene 6000 v1.7 software.

**Table 1. Transcript selection and primer design summary.** Size refers to qPCR amplicon; efficiency was calculated with a serial dilution of 4-fold.

| Transcript | Name | Sequence (5'—3') | Size (bp) | Efficiency | $R^2$ | GenBank |
|---|---|---|---|---|---|---|
| LvRPL7 | Ribosomal protein L7 | GCC GTG TTC CTC TGT ACT CC | 161 | 2.01 | 0.98 | JX481271.1 |
| | | GAG TTC AAA GCC GTC AAT ATC C | | | | |
| LvPEPCK | Phosphoenolpyruvate carboxykinase | GGA GAG CAA GAC CTT CAT CG | 153 | 2.1 | 0.98 | AJ250829.1 |
| | | GTT CTT CCC TTC ATG CAT CC | | | | |
| LvTHBS | Thrombospondin | CCA CAA ACG GCA TCT ACT CC | 201 | 1.93 | 0.98 | MK033611 |
| | | ATA GAA CTT GGC ATT GCT TTG G | | | | |
| LvPRDX | Peroxiredoxin | TCA ACG CGG TCA TTA AAG G | 170 | 1.95 | 0.97 | JN393011.1 |
| | | AAA GCC ATG ATC AGC AAA CC | | | | |
| LvCAS | Cellular apoptosis susceptibility protein | TGT GCA GTT GGT CTC AGT GG | 183 | 1.97 | 0.95 | MK033612 |
| | | TCT GGC TGA TGA ACC TTG G | | | | |
| LvCLO | Cottable protein | AAG GTC ATA AAG CAG GGT AGT CG | 112 | 2.14 | 0.99 | DQ984182.1 |
| | | GAC CGT GAG GAC AGA GAA GC | | | | |
| LvHCY | Hemocyanin | CAA CAT CGT CCA CAT CTT CG | 159 | 1.91 | 0.99 | KY695246.1 |
| | | TTG ATG CTG AAC GTC TGT CC | | | | |
| LvcSOD | Superoxide dismutase | GCC CCA ATG AGA ACA AGC | 146 | 1.80 | 0.98 | DQ005531.1 |
| | | AAG GCC TTC ACG TAA TCT GC | | | | |
| LvHYO | Hypoxia up-regulated gene | ATG TTT TTG CCA AGC TCT CG | 121 | 1.97 | 0.95 | MK033613 |
| | | CAC TGA TTT CTG CTG CTA TGG | | | | |
| LvCTR | Chymotrypsin | TCT TCA CTC ACG AGC ACT GG | 192 | 1.89 | 0.98 | X66415.1 |
| | | GGA GGA CGT CGG AGA TAC C | | | | |
| LvCTSH | Cathepsin L | TCA TCG ACG ACC ACA ACG | 84 | 1.92 | 0.96 | X85127.1 |

In all cases, qPCRs were carried out in triplicate, in a Rotor-Gene 6000 real-time rotary analyzer (Corbett Research, Sydney, Australia). The qPCR reactions contained the following final concentrations: 2.5 mM MgCl2, 0.2 mM dNTPs (each), 1.25 U of GoTaq Flexi DNA Polymerase (Promega, Madison, WI), 500 nM of each primer, 1X EvaGreen Dye (Biotium, Hayward, CA), and ~17 ng of cDNA 1:50 diluted in 15 μL of final volume. The amplification conditions were: 95 ˚C (5 min), 40 cycles of 95 ˚C (20 s), 60 ˚C (20 s) and 72 ˚C (20 s) with a final extension of 72 ˚C (5 min). The melting curve amplification was performed from 55 ˚C to 99 ˚C (1 ˚C/s).

Based on the $2^{-\Delta\Delta Cq}$ calculation, the qPCR data was analyzed using a one-way ANOVA; significant statistical differences were obtained with Holm-Sidak multiple comparisons test ($\alpha = 0.05$). Final statistical analyses were performed in SigmaPlot v11 (Systat Software Inc., Germany). Statistically significant differences were set at $p < 0.05$ [53].

## Results

### Characterization of *V. parahaemolyticus* IPNGS16 strain

To confirm that we were using the correct bacterial strain for our infection experiments, an end-point PCR was performed to screen for the *PHP* and *PirB* gene sequences from two different *V. parahaemolyticus* strains: the IPNGS16 strain which tested positive for the two gene sequences and the CAIM170, a non-AHPND causing strain which is negative for *PHP* and *PirB* sequences (S1 Fig).

### Experimental infection with *V. parahaemolyticus* IPNGS16 and lethal concentration 50 (LC₅₀)

To determine the $LC_{50}$, first, we tested a range of lethal and sub-lethal bacteria concentrations going from 3,000 to 350 CFU/mL. We observed that using 3,000 CFU/mL, 60% of the

specimens reached mortality by 12 hpi and by 24 hpi, mortality had reached 100%. With concentrations of 1,500 and 1,000 CFU/mL, specimens showed 75–80% mortality by 24 hpi, and by 48 hpi, 100% mortality was reached in both concentrations. Finally, using concentrations of 750 and 350 CFU/mL, specimens reached 35–40% mortality by 24 hpi. By 48 hpi, mortality reached between 80 to 90%, and by 72 hpi, mortality was 100% (Fig 1A). Using the values obtained in this experiment, a Probit analysis was run to calculate the $LC_{50}$. The $LC_{50}$ value obtained was equivalent to 660.95 CFU/mL at 24 hpi (Fig 1B).

## Histopathology of hepatopancreas

Early stages of AHPND in infected shrimps are characterized by shedding of epithelial cells in the hepatopancreas, followed by necrosis of epithelial cells, and hemocytic infiltration at later stages [6]. To confirm that our specimens were indeed developing AHPND due to *V. parahaemolyticus* infection, we selected 5 infected and 5 control organisms at 0, 12, 24, 48, and 72 hours to perform histological sections, and hematoxylin and eosin staining. The hepatopancreas epithelium is distinguished for having well defined and clear tubule structures (Tub), a star shape lumen (Lum), and B-cells (HpB) (Fig 2A).

Our observations indicate that by 12 hpi, there is a sloughing (Slo) of epithelial cells from the tubule (Fig 2B). At 24 hpi, epithelial tissue shows a clear presence of pyknotic cells, a characteristic feature for necrosis (Nec) (Fig 2C). By 48 hpi, the tubule lumen (Lum) has lost its distinctive star-shape (Fig 2D) and, by 72 hpi, structural atrophy (Atr) and epithelial cell detachment is seen in B, R, and F cells, which are structural components of the hepatopancreas tubule (Fig 2E), and granular hemocytes (HemG) and hyaline hemocytes (HemH) were observed (Fig 2F).

## *De novo* assembly and functional annotation

Two cDNA libraries were constructed from non-infected and infected shrimp hepatopancreas, the total raw data obtained from both libraries were 166,174,430 paired-end reads.

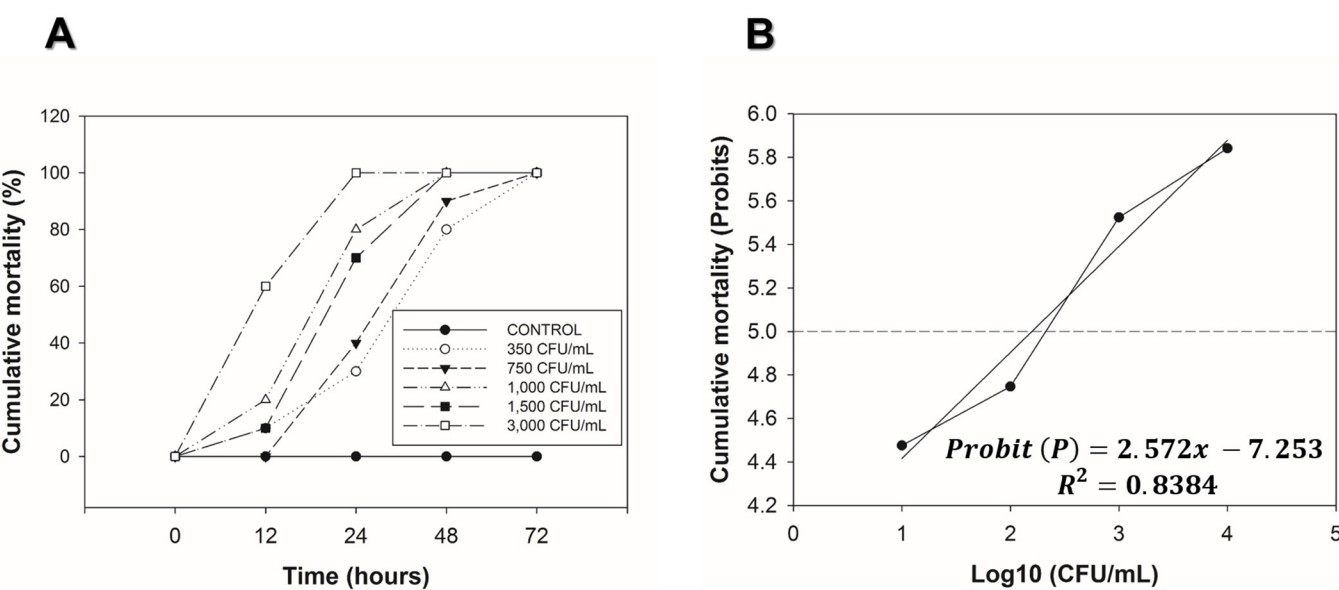

**Fig 1. $LC_{50}$ doses calculation at different time points in *Litopenaeus vannamei* after *Vibrio parahaemolyticus* IPNGS16 infection.** (A) $LC_{50}$ standardization curve using a range of bacterial concentrations (from 350 to 3,000 CFU/mL), X-axis shows different time points where shrimp mortality was measured, Y- axis represents the percentage of cumulative shrimp mortality (n = 30). (B) The graph represents a Probit model, in the X- axis are the bacteria concentrations values [CFU/mL], in the Y- axis the Probit values for cumulative mortality, the dotted line represents the value of intersection to calculate the $LC_{50}$. The $LC_{50}$ value was determinate at [660.95 CFU/mL].

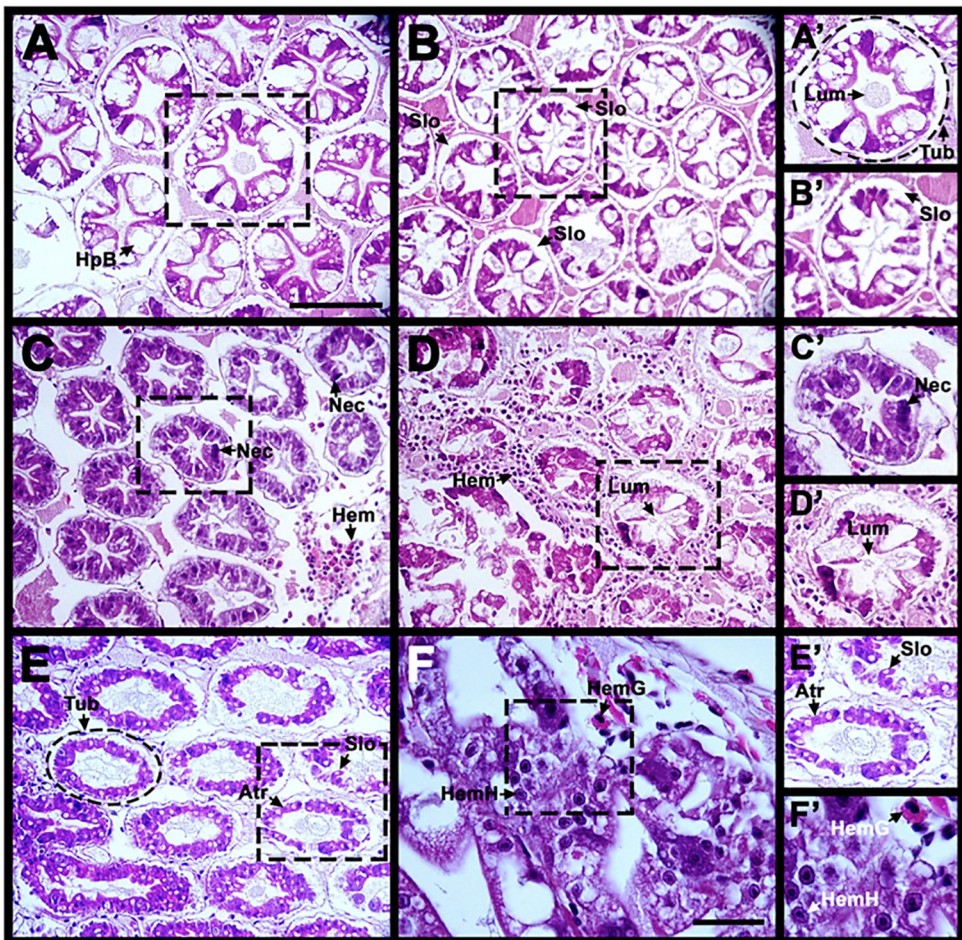

**Fig 2. Light micrographs of longitudinal sections of hepatopancreas (Hp) from *Litopenaeus vannamei* infected with *Vibrio parahaemolyticus* IPNGS16.** Sections were stained with Hematoxylin-eosin, five samples were analyzed for each different time point. Untreated sample at 0 h. Normal Hp has clear tubules (Tub,) (A'), a star-shaped lumen (Lum) (A'), and B-cells (HpB). B- F). Infected samples at different time point post-infection. B,-B'). Small spaces can be seen between epithelial cells indicating sloughing cells (Slo) at 12 hpi. C- C') Necrosis (Nec), and hemocyte infiltration (Hem) is present in the tubular epithelium, at 24 hpi. D) Presence of hemocyte infiltration is accompanied by lumen (D') and epithelium reduction at 24 and 48 hpi. E- E') Atrophy in epithelial cells (Atr) and sloughing cells in tubules is seen at 72 hpi. F) 100× magnification to visualize granular (HemG) and hyaline (HemH) hemocyte infiltration. A'-F') are magnified views of white dotted rectangle regions in panels (A- F) respectively. Scale bar: 100 μm (A–E) and 30 μm (F).

After sequence trimming, a total of 148,571,949 paired-end reads were retrieved from both libraries. Trimmed paired-end reads obtained from the raw data were deposited in the Short Read Archive (SRA) database of The Nacional Center for Biotechnology Information (NCBI) with the following SRA accession codes SRR7986779 (non-infected), and SRR7986780 (infected). According to MIGS standards, the summary of this project is shown in S1 Table [54].

After the *de Novo* assembly was completed, we found 174,098 transcripts with an $N_{50}$ value of 1,538 bp, and an average transcript mean size of 984.07 bp (Table 2), of which 86,388 transcripts (49.6%) were 450bp in length, followed by 34,272 transcripts (19.8%) around 900 bp and 12,375 transcripts were over 3 Kb (Fig 3B). All transcripts obtained were annotated with BLASTx using SwissProt (96,988 hits) and UniProtKB$^{INV}$ (199,139 hits). For BLASTp we only

**Table 2. Summary of assembly and annotation.** Data from the hepatopancreas of *Litopenaeus vannamei* with AHPND.

| Analysis | Number |
|---|---|
| Number of contigs | 174,098 |
| GC content (%) | 43.36 |
| Transcript $N_{10}$ | 1,961 |
| Transcript $N_{50}$ | 1,538 |
| Longest transcript (nt) | 16,189 |
| Shortest transcript (nt) | 201 |
| Median transcript length | 455 |
| Average transcript length | 984.07 |
| > 10 kb | 104 |
| Greater than 5 kb | 3,484 |
| Greater than 3 kb | 12,385 |
| > 1 kb | 49,640 |
| Total assembled bases | 171,324,032 |
| Total transcripts with ORF | 149,723 |
| Longest ORF (aa) | 5,141 |
| Average ORF length (aa) | 100 |
| Annotation of all transcripts | 96,988 |
| Annotations of predicted proteins (ORF) | 41,807 |
| Predicted protein with Pfam domain | 39,987 |
| Signal peptide | 10,586 |
| Transmembranal regions | 32,365 |
| EggNOG hit | 39,279 |
| KEGG hit | 40,883 |
| GO Annotation | 47,886 |
| GO annotation (based in Pfam) | 25,472 |

used SwissProt (41,807 hits). Sequence identity percentage and e-value distribution histograms are shown in Fig 3C and 3D, respectively.

Gene ontology (GO) annotations were assigned to 47,886 transcripts (Table 2), the SwissProt database was only used to assign annotations. The three main ontologies (Fig 4A) represented are: 41.0% biological process (BP), 32.86% cellular components (CC), and 26.24% molecular functions (MF). The COG database was used to generate orthologous protein groups from the assembled transcripts. Functional categories were identified (Fig 4B), being the translational ribosomal structure and biogenesis categories with the highest number of associated transcripts (14.75%), followed by general function prediction only (12.54%), amino acid transport and metabolism (7.05%), energy production and conversion (6.64%), and carbohydrate transport and metabolism (5.74%). The transcripts obtained were annotated using the KEGG database to determinate the relationship between transcript function and their specific metabolic pathway (Fig 4C). A total of 8,083 transcripts were classified in metabolic pathways; within this category, 1,191 transcripts were in carbohydrate metabolism and 1,279 transcripts related to the family of enzymes (Fig 4C). Interestingly, the category that contains the highest number of transcripts (12,943) is the processing of genetic information, which includes genes for zinc finger protein, transcription factors, and basic loop-helix, a second group of 4,049 transcripts were classified into the folding, sorting and protein (Fig 4C).

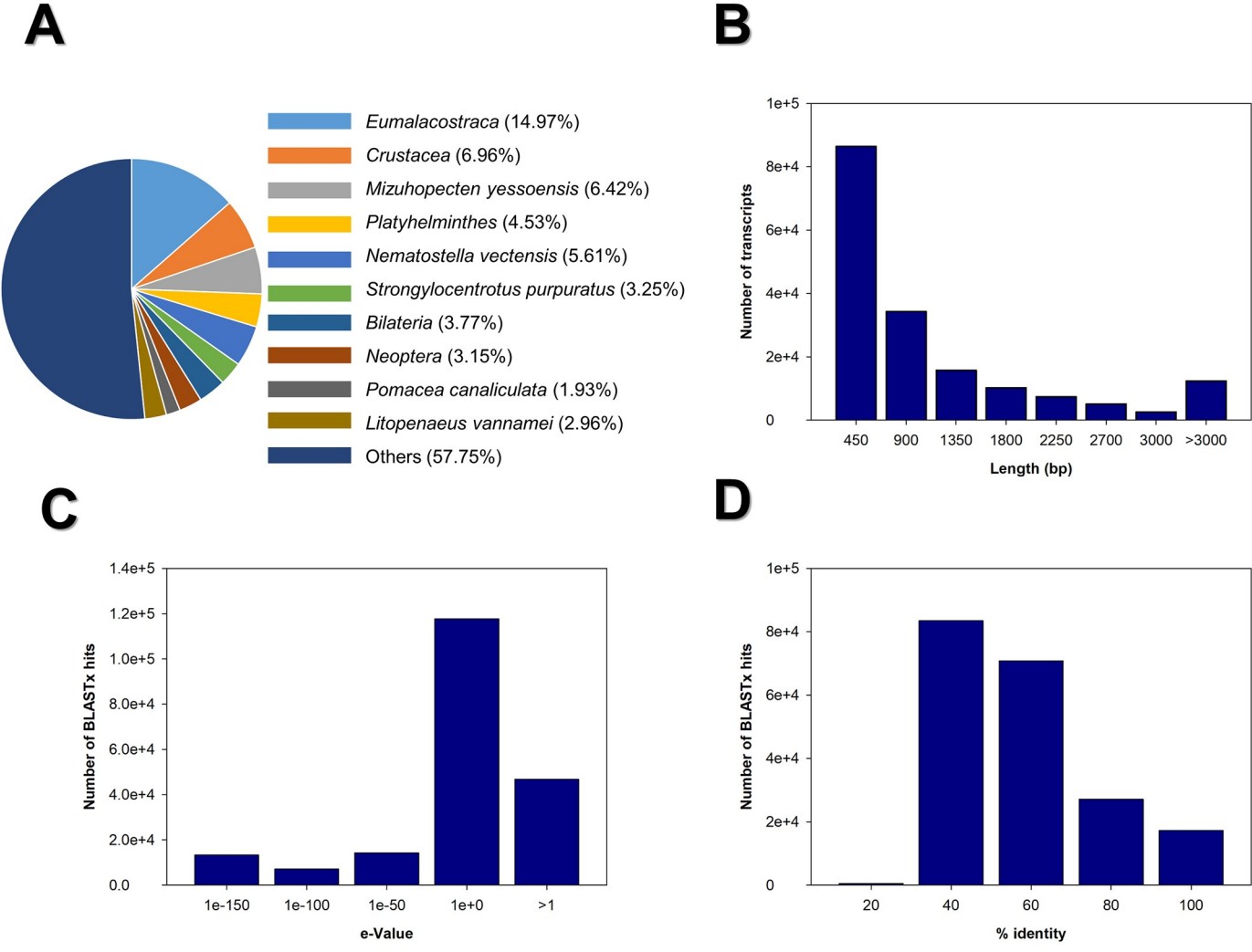

**Fig 3. BLASTx statistics, transcript length distribution e- value probability and percentage of transcript identity.** RNAseq data was validated using the following parameters: A) Species distribution of the BLASTx using Swissprot and UniProtKB and filtered with the UniProt^INV database which includes the invertebrate phyla: Arthropoda, Mollusca, Porifera, Cnidaria, Echinodermata, Platyhelminthes, Nematoda and Annelida. B) Transcript length distribution after trinity assembly. In the X- axis the total transcript set was grouped into different lengths (shown in base pairs, bp), the Y- axis shows the frequency of each transcript group. C) An e-value was calculated (X- axis) to determine the number of random hits in our BLASTx (Y- axis). D) Percentage of identity (in the X- axis) was also calculated after the BLASTx, shown in the Y- axis as number of BLASTx hits.

## Determination of differentially expressed transcripts (DETs) in hepatopancreas

Based on the FDR≤0.01 and Log2Ratio≥2 threshold, a total of 915 transcripts were identified as DETs, from which two clusters were obtained, one containing 442 up-regulated transcripts and a second cluster with 473 down-regulated transcripts (Fig 5). Both clusters were annotated with BLASTx and BLASTp using the SwissProt and UniRef90 databases, in S2 and S3 Tables show a list of the up and down-regulated transcripts. To further analyze the functional role of these transcripts, an enrichment analysis was performed to obtain enriched and depleted GO terms in the up and down-regulated transcripts and grouped into three main categories: biological process, molecular function and cellular components (Fig 6A and 6B). Up-regulated transcripts show the following enriched terms for biological process: metabolic process (7,015

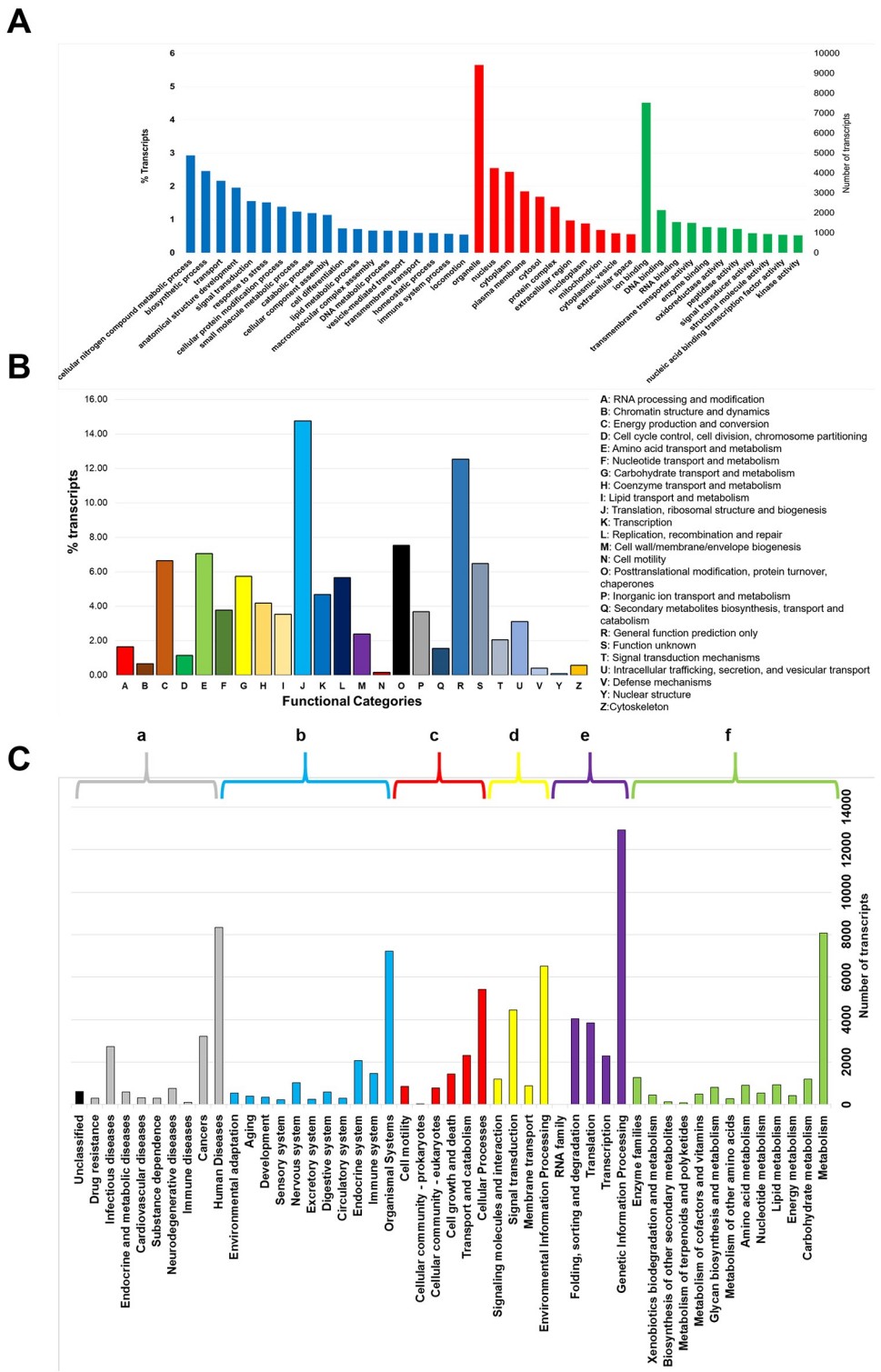

**Fig 4. Annotation of *Litopenaeus vannamei* transcripts.** Annotated transcripts were classified separately using three database resources: (A) Gene ontology (GO), transcripts were grouped in three different ontologies: biological process (BP) in blue, cellular component (CC) in red, and molecular function (MF) in green. The X- axis shows each GO ontology and all the biological categories that lie in each ontology. The Y- axis in the left hand side (LHS) shows the percentage of transcripts (%) for each individual category and the Y- axis in the right hand side (RHS) the number of transcripts in each category (B) The second functional classification was using the orthologous cluster group (COG). The X- axis represents the protein categories predicted which are listed in the RHS. (C) In the third classification, six

categories were assigned using the Kyoto Encyclopedia of Genes and Genomes (KEGG) listed top of the chart: (a) Cellular processes, (b) Environmental Information Processing, (c) Genetic Information Processing, (d) Human Diseases, (e) Metabolism, and (f) Organismal Systems. In the X- axis are the gene function assignations for each category. On the Y- axis in the RHS are the number of transcripts found for each gene assignation.

transcripts), regulation of programmed cell death (593 transcripts) and regulation of apoptotic process (570 transcripts). For molecular function the enriched terms were: carbohydrate derivative binding (1,875 transcripts), oxidoreductase activity (1,063 transcripts) and transition metal ion binding (908 transcripts). The GO terms for cellular components were: extracellular region part (1,694 transcripts), vesicle (1,566 transcripts) and extracellular organelle (1,149 transcripts) (Fig 6A). Enriched terms for the down-regulated transcripts showed the following terms, for biological process: carbohydrate metabolic process (576 transcripts) and microtubule-based process (359 transcripts); for molecular function: binding (8,373 transcripts) and catalytic activity (5,471 transcripts); for cellular component: extracellular part (1,694 transcripts) and extracellular region (896 transcripts) (Fig 6B).

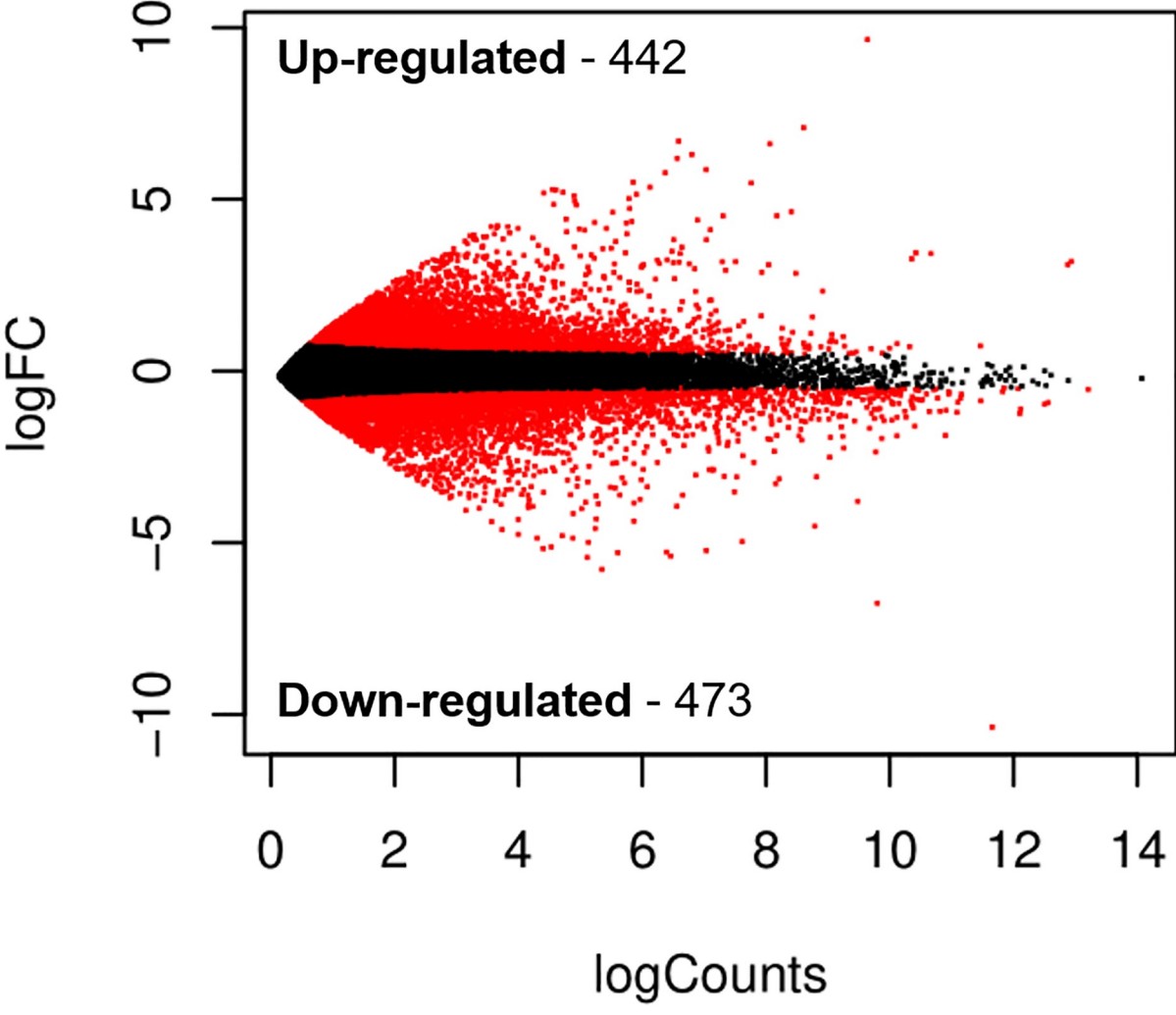

**Fig 5. Differentially expressed transcripts.** Volcano plot shows the DETs from shrimp hepatopancreas challenged with *Vibrio parahaemolyticus* 16IPNIAV13 at 24 hpi. Log counts in the X- axis indicate transcript log-abundance. LogFC in the Y- axis indicates fold change; each dot represents one gene; red dots are either up or down-regulated genes and black dots are genes with no differential expression.

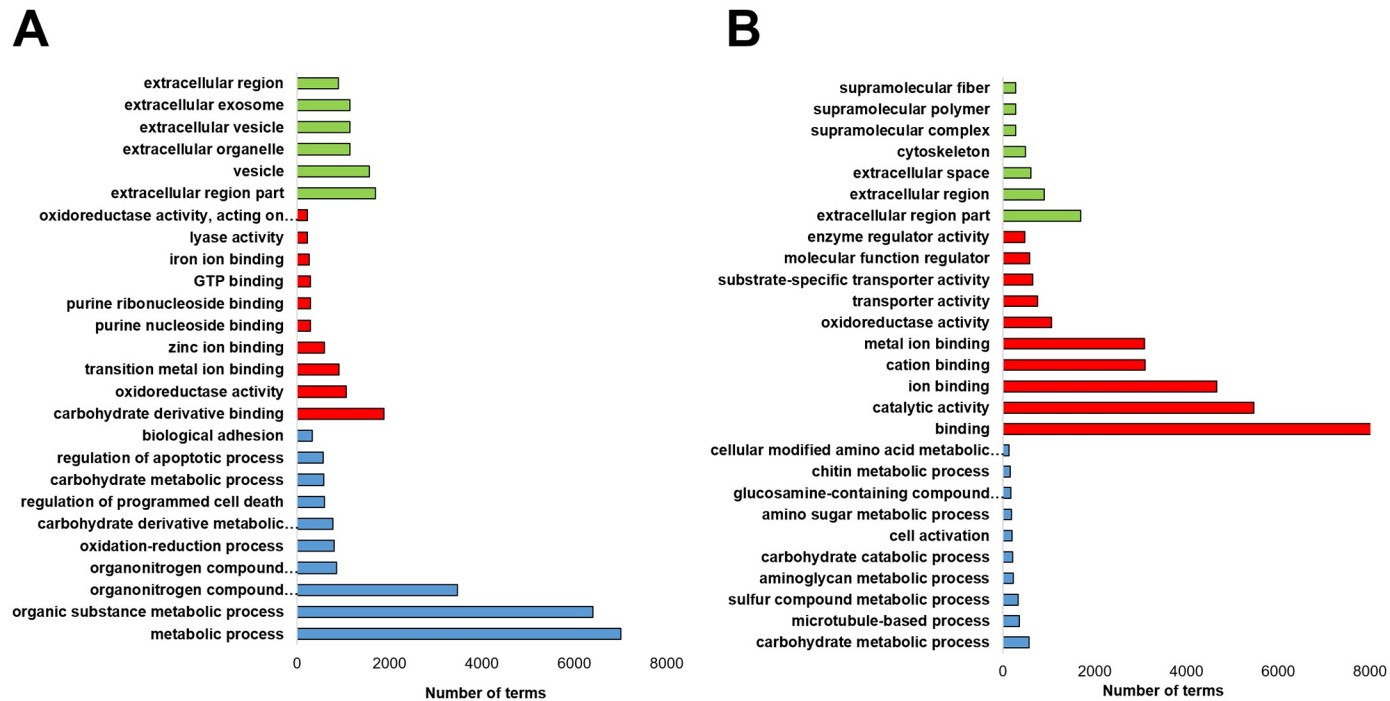

**Fig 6. DETs Gene Ontology (GO).** DETs were grouped in three main GO categories: biological process (blue bars), molecular function (red bars), cellular component (green bars). A) Up-regulated transcripts. B) Down-regulated transcripts. The X-axis in both charts indicates the number of enriched terms ($P<0.01$).

## Differentially expressed transcripts at protein-protein analysis

The DETs found in the assembled hepatopancreas transcriptome were analyzed in GeneMA-NIA using *Drosophila melanogaster* as source species to predict protein-protein interactions between up and down-regulated transcripts. We found 11 protein-protein interactions predicted by the GeneMANIA algorithm. Based on their biological functions, we constructed three regulatory networks, the first network (*Network 1*) represents up-regulated genes (S2A Fig), it was constructed with 203 predicted proteins, the main enriched terms are chitin metabolic process, which represents 3.94% of the proteins in the network, homophilic cell adhesion via plasma membrane adhesion molecules (2.46%), and oxidation-reduction process (7.88%) see in S4 Table. *Network 2* represents down-regulated transcripts (S2B Fig), 230 predicted proteins were used to construct this network, the enriched terms are axonogenesis (3.083%), sarcomere organization (2.20%), and oxidation-reduction process (7.48%) see in S5 Table. Lastly, we constructed a third network combining the up and down-regulated transcripts (*Network 3*), the enriched terms associated to this network are oxidation-reduction process (7.63%), sarcomere organization (1.84%), and imaginal disc-derived wing morphogenesis (4.73%) (S6 Table). Network 1 contains 203 nodes and 2,264 edges, Network 2 has 230 nodes and 3,352 edges, and Network 3, has 383 nodes and 8,557 edges (S2 Fig).

## Identification of hub transcripts

One of our most significant findings come from *Network 3*, it contains 20 hub genes that represent the maximum number of interactions within any of the three networks (Table 3). One of the main hub genes identified in *Network 3* was *thrombospondin*, the GO terms assigned to this gene are: wound repair, cell adhesion and immune system response. Interestingly, when *thrombospondin* is up-regulated, the enriched GO terms are associated with anatomical structure

**Table 3. Top 20 of hub genes.** *Network 3* predicted Hub genes and their homologs.

| UniProt ID | Homolog protein (*Drosophila melanogaster*) | GO Term (biological process) | Homolog protein in UniRef90 (specie) |
|---|---|---|---|
| Q9VZI1 | Transgelin | juvenile hormone mediated signaling pathway | Calponin-3 (*Mustela putorius furo*) |
| Q95RI5 | Failed axon connections | axonogenesis | Failed axon connections homolog (*Xenopus tropicalis*) |
| P31409 | V-type proton ATPase subunit B | ATP hydrolysis coupled proton transport | ATP synthase subunit alpha, mitochondrial (*Drosophila melanogaster*) |
| Q7KIF8 | Ran binding protein 7 | Cajal body organization | Cellular apoptosis susceptibility protein (*Fenneropenaeus chinensis*) |
| Q7K3N4 | CG8888 | | Uncharacterized protein (*Scylla olivacea*) |
| M9PI82 | Mo25, isoform B | | Protein Mo25 (*Drosophila melanogaster*) |
| Q8MRJ0 | LD15094p | acylglycerol transport | Apolipophorins (*Locusta migratoria*) |
| A1Z8M2 | Toutatis, isoform A | ATP-dependent chromatin remodeling | Uncharacterized protein (*Scylla olivacea*) |
| Q7JRF0 | RE48511p | | UDP-glucuronosyltransferase 2B14-like (*Hyalella azteca*) |
| O46067 | CG2918, isoform A | multicellular organism reproduction | Hypoxia up-regulated protein 1 (*Xenopus laevis*) |
| A0A0B4JD46 | CG3800, isoform B | | Thrombospondin (*Marsupenaeus japonicus*) |
| Q9W0I2 | RE15268p | adult chitin-containing cuticle pigmentation | C-type lectin 1 (*Scylla paramamosain*) |
| Q961F2 | GH27269p | | Uncharacterized protein (*Scylla olivacea*) |
| Q8IRD3 | Glutathione peroxidase | response to oxidative stress | Glutathione peroxidase (*Procambarus clarkii*) |
| Q7K0F7 | Carbonyl reductase, isoform A | | Uncharacterized protein (*Scylla olivacea*) |
| Q9VP77 | LD23875p | | dnaJ homolog subfamily C member 2-like (*Hyalella azteca*) |
| E1JHR5 | Enolase, isoform F | glycolytic process | Uncharacterized protein (*Fundulus heteroclitus*) |
| A2VEJ9 | CG4266, isoform C | | Uncharacterized protein (*Plasmodium gonderi*) |
| Q7KN75 | Dodeca-satellite-binding protein 1, isoform A | chromosome condensation | Vigilin (*Homo sapiens*) |
| Q9VSC9 | CG7565 | | Uncharacterized protein (*Scylla olivacea*) |

morphogenesis, localization, multi-organism process, cell development, and stress response. Whereas down-regulated thrombospondin enriched GO terms are associated with intercellular transport, male germ-line stem cell population maintenance, protein N-linked glycosylation, neuron projection morphogenesis, and positive regulation of JAK-STAT cascade (Fig 7).

## Gene selection, relative gene expression analysis and transcriptome validation

In order to validate candidate transcripts, we selected ten transcripts based on the following criteria: the GO enrichment analysis data, the fold-change value obtained for each transcript and their known function (S4 and S5 Tables). The main enriched GO terms were in oxidation-

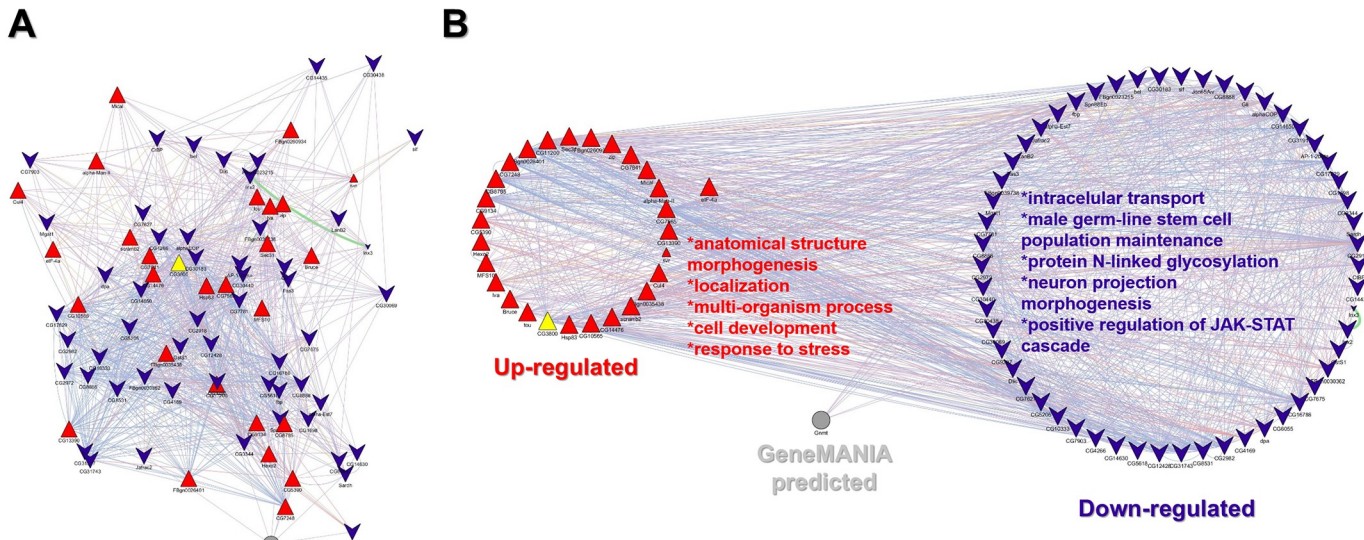

**Fig 7. Thrombospondin gene interacting network.** A) Topology of the hub thrombospondin (yellow triangle) network with its first up and down -regulated gene neighbors. B) Circular interacting gene network, thrombospondin is in the unregulated group. In both networks, functional enriched up-regulated transcripts are in red triangles, down-regulated transcripts are in blue chevrons, the gray circle represents a set of non-immunological related genes predicted by a GeneMANIA.

reduction processes, carbohydrate derivative metabolic processes, apoptosis regulation and biological adhesion (Fig 6A), the selected genes are shown in Table 1 [55].

The RNA-seq expression results were validated by RT-qPCR using the original extracted RNA from both control and infected hepatopancreas samples from different time points after infection (S3 Fig).

Based on their relative expression, DETs were grouped into two categories, those related to the immune system (Fig 8) and metabolic processes (Fig 9). *LvSOD* (Fig 8A) increased its expression at 6, 24, 48 hpi in the infected group compare to controls. However, a significant decreased in expression is observed at 12 hpi. *LvPRDX* showed a gradual increased expression from 12 to 48 h in the infected group (Fig 8B). Relative expression of the *LvCLO* gene, which encodes for a coagulation protein, increased significantly only at 3 and 24 hpi (Fig 8C). The *LvHYC* gene that encodes for hemocyanin increased its expression by 150-fold at 24 hpi and decreases by 48 h, although there are not significant statistical differences between the two time points (Fig 8D). Expression of *LvTHBS* that encodes for *thrombospondin* had a slight increase in its expression at 3, and 48 hpi but, a decrease in expression at 6, and 24 hpi (Fig 8E). Lastly, the cellular apoptosis susceptibility gene *LvCAS*, showed increased expression at 3, 12, 24 and 48 hpi when compare to the control group, reaching its highest expression (15 fold) at 24 hpi (Fig 8F).

The relative expression of genes connected to metabolism is shown in Fig 9. The phosphoenolpyruvate carboxykinase (*LvPEPCK*) gene starts to increase its expression by 6 and 12 hpi and by 24 hpi its expression increased 20-fold in comparison to the control group (Fig 9A). The chymotrypsin gene (*LvCTR*) has small increments in its expression at 3, 6 and 24hpi; however, the only statistically significant difference in increase expression is at 24 hpi when compared to the control group (Fig 9B). Gene expression of hypoxia-up-regulated gene (*LvHYO*) increased at 6 and 24 hpi; its expression was decreased at 12 and 48 hpi (Fig 9C). Finally, the *LvCTSH* expression, a gene which encodes for the Cathepsin L protein, showed a significant decrease at 24 hpi (Fig 9D), the rest of the time points were not statistically significantly difference compared to the controls.

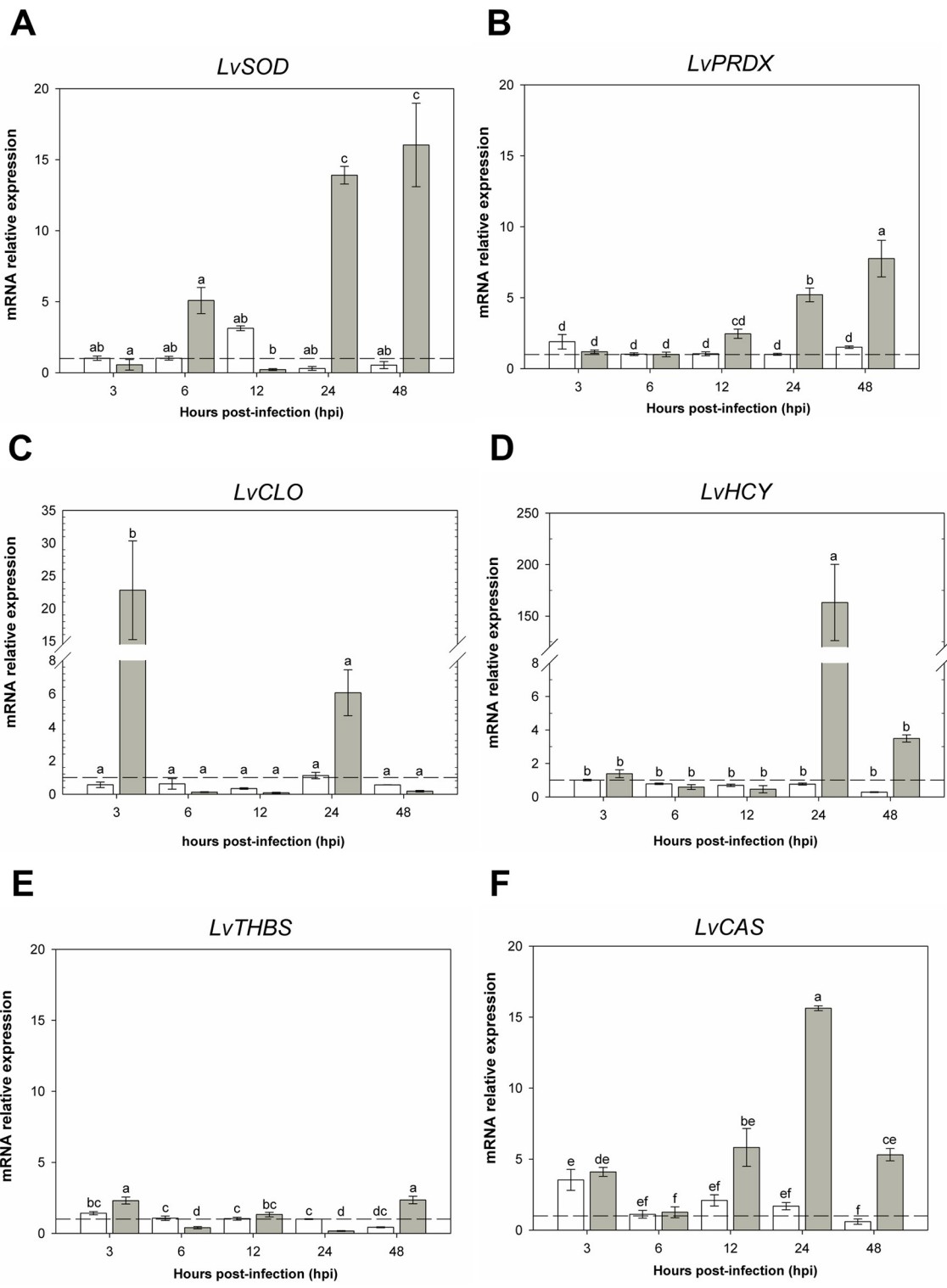

**Fig 8. mRNA expression of immune system- related genes.** RT-qPCR results showing the relative mRNA expression of DETs. For all the charts, the X-axis shows different time point post-infection (shown as hpi), the Y-axis shows relative mRNA expression (target gene/*LvRPL7*); bars show the mean and error lines the standard error (n = 9). Dash line represents the normalized value and lower case letter(s) show statistically significant differences (p < 0.05). A) Cytosolic superoxide dismutase. B) Peroxiredoxin. C) Cottable Protein. D) Hemocyanin. E) Thrombospondin. F) Cellular apoptosis susceptibility protein.

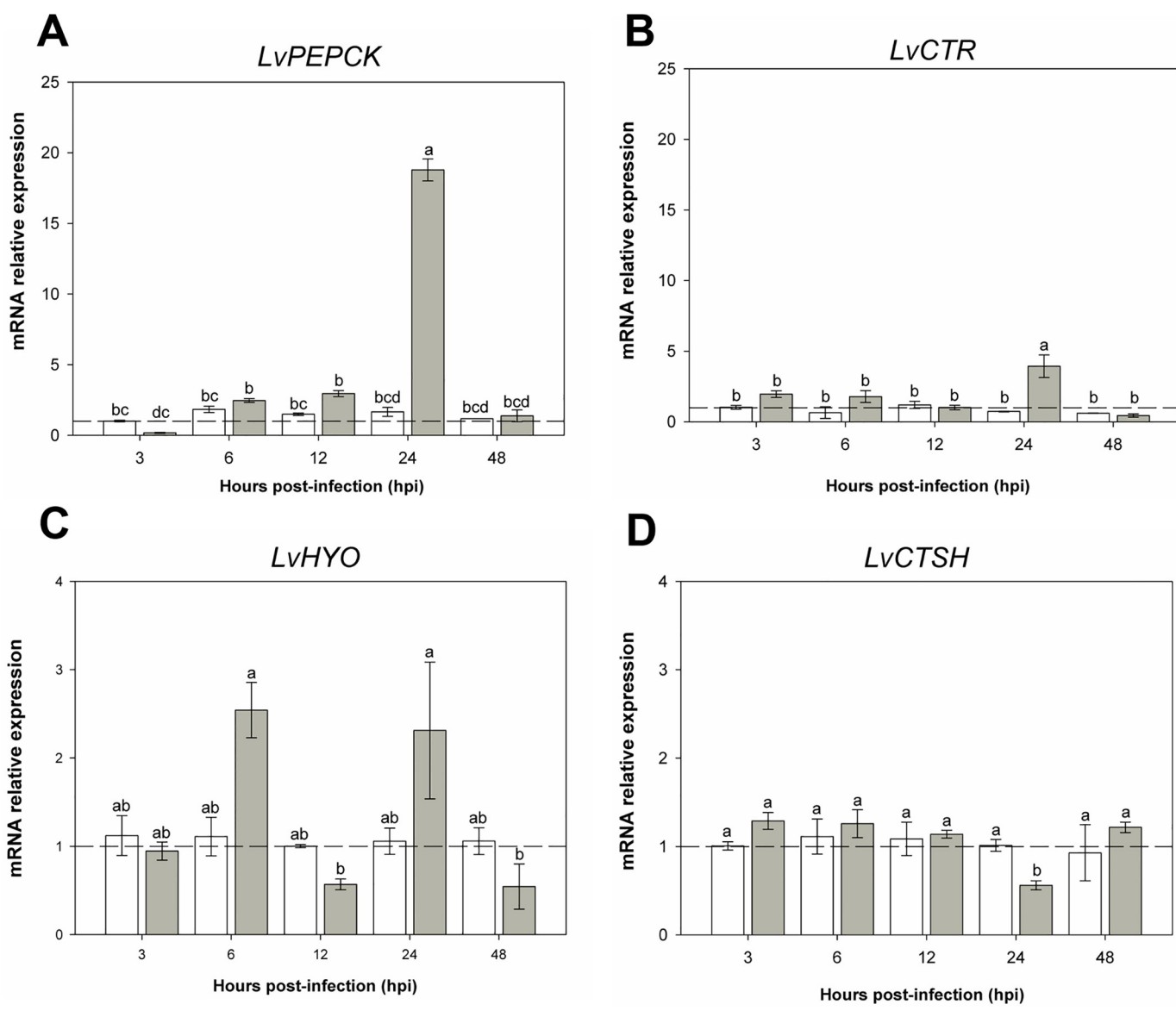

**Fig 9. mRNA expression of metabolic process- related genes.** RT-qPCR results showing the relative mRNA expression of DETs. For all the charts, the X-axis shows different time point post-infection (shown as hpi), the Y-axis shows relative mRNA expression (target gene/*LvRPL7*); bars show the mean and error lines the standard error (n = 9). Dash line represents the normalized value and lower case letter(s) show statistically significant differences (p < 0.05). A) Phosphoenolpyruvate carboxykinase. B) Chymotrypsin. C) Hypoxia up-regulated protein. D) Cathepsin L protein.

## Discussion

Acute hepatopancreatic necrosis disease (AHPND) is an emergent disease caused mainly by *V. parahaemolyticus*, which induces 90–100% mortality in shrimp cultures worldwide [56]. In this study, we have identified a significant number of DETs and started to understand the molecular pathways that take place during AHPND. Before the experimental infection trials, the *V. parahaemolyticus* strain was characterized by PCR assay by amplifying the *PirB* and *PHP* genes. A LC$_{50}$ was standardized and set at 660 CFU/mL, this concentration represents 50% mortality at 24 hpi, and these data were used for our subsequent RNA-seq experiment.

Juvenile *L. vannamei* were infected with *V. parahaemolyticus* IPNGS16 (AHPND positive) by immersion [57]. Following histopathological results at 12, 24, 48 and 72 hpi a sequential progression of hepatopancreas lesions was observed in AHPND infected tissue. Necrosis, epithelial cells sloughing, atrophy, and massive hemocyte infiltration signs were more evident from 48 hpi onwards (Fig 2D), our histopathological results agree with previous reports [4,6,57].

In our differential expression analysis, a total of 915 transcripts were detected between the infected and non-infected control groups. The number of up-regulated genes was considerable more (473) than the down-regulated genes (442); probably due to biological changes and inhibition of basal cellular functions caused by *V. parahaemolyticus* infection [58]. Our knowledge of the AHPND pathogenesis indicates that *V. parahaemolyticus* produces toxins that cause cell sloughing [59], interestingly we found two immune- and metabolic-related gene clusters in our KEGG analysis. The immune-related gene cluster contains antimicrobial peptides (AMPs), oxidative stress proteinases and inhibitory proteinases, pattern recognition, coagulation, and signaling pathway proteins, along with other immune-related proteins (S1 and S2 Tables). In the metabolic-related cluster, we found genes related to carbohydrate metabolism, beta-oxidation, lipid metabolic processes, and fatty acid biosynthesis (S1 and S2 Tables). This differential transcript analysis provides us with the basic information about the genes, their expression and the immune and metabolic processes that take place during AHPND [60].

The *L. vannamei* innate immune system contains cellular and humoral components that work individually and synergistically to protect the organism integrity during infection. Activation of the shrimp immune response depends on diverse mechanisms that include the antioxidant system, hemolymph, wound repair and apoptosis-related proteins [60].

All living organisms produce the reactive oxygen species (ROS) during normal aerobic metabolism. Stress conditions like pH, temperature, hypoxia and microbial infections can shortage oxygen and increase ROS, resulting in oxidative stress within the cells [61]. Increased levels of ROS causes macromolecules damage which in turn affect membranes and enzymes structure and nucleic acids integrity [62]. The endogenous cellular defense mechanisms to stabilize increased levels of ROS include antioxidant molecules and enzymes (catalase, glutathione peroxidase, and superoxide dismutase) [63]. Our GO enriched analysis ($P < 0.01$) show 442 up-regulated transcripts which were grouped in different categories including oxidation-reduction processes, biological process (BP) and oxidoreductase activity for molecular function (MF).

In this work, we identify two oxidative stress-related transcripts, peroxiredoxin (up-regulated) and superoxide dismutase (down-regulated). Peroxiredoxin belongs to a group of antioxidant enzymes that protect the cell from ROS by reduction of a wide range of cellular peroxides [64]. Our gene expression analysis results show that *LvPRDX* expression increases significantly after 12 hpi (Fig 8B). This result agrees with a previous report where peroxiredoxin 5 expression levels increase during *V. angillarum* infection in *E. carinacauda* [65]. We also found that the antioxidant superoxide dismutase (*LvSOD*) gene was down-regulated in our RNA-seq data (S3 Fig), however in our qPCR results *LvSOD* expression was increased at 6, 24, and 48 hpi (Fig 8A). Although we are not certain of why *LvSOD* shows a complete difference in expression, we know that *V. parahaemolyticus* and white spot syndrome virus (WSSV) infections increase mRNA levels of these genes, and ROS play an essential role in *L. vannamei* defense against these pathogens [66].

Hemocyanin, an abundant protein present in the hemolymph of arthropods and mollusk [67]. Its primarily function is to transport and store molecular oxygen, along with other multiple physiological functions like protein storage, osmoregulation, molt cycle (post-molt (A) and pre-molt (D)), exoskeleton formations, and antimicrobial activity as well as a non-specific

immune molecule [68,69]. Our RNA-seq data shows that multiple isoforms of *LvHCY* are present in both up and down-regulated transcript sets. Gene expression analysis of *LvHCY* shows a significant increase expression after 24 hpi compare to the control. Previous studies have shown that Hemocyanin is overexpressed during AHPND and PirAB challenge. Furthermore, Boonchuen and colleagues demonstrated the toxin-neutralizing activity of hemocyanin against *V. parahaemolyticus* infection [70].

Among of the first immune responses that have the organisms to either fight against pathogens or to prevent loss of hemolymph during tissue damage is clotting.

Our results indicate that hemolymph cottable protein (*LvCLO*) expression increases significantly at 3 hpi studies have shown that disrupting the ProPO system, a coagulation-related cascade, using miRNA silencing increases bacterial load in the host [71]. Moreover, genomic studies in *P. monodon* showed up-regulation of coagulation-related genes during stomach infection caused by *V. parahaemolyticus* [12]. These findings show activation of the coagulation system is a fundamental step during bacterial infection [72].

Thrombospondins (THBSs) are a family of extracellular calcium-binding glycoproteins, in vertebrates THBSs are involved in cell-extracellular matrix interactions, angiogenesis, synaptogenesis and organization of connective tissue [73]. However, very little is known about the function of THBSs in invertebrates, for example in penaeid shrimp, it has been suggested that THBSs are part of the defense response against microbial infection, in *Fennerpenaeus chinenesis* challenged with *V. angillarum* and *Staphylococcus aureus*, TSP is up-regulated in hemocytes, heart, intestine, stomach, and hepatopancreas [74]. Moreover, in *P. monodon* THBSs are implicated in the immune function of the lymphoid organ against two shrimp pathogens, *V. harveyi* and the white spot syndrome virus (WSSV) [75] and, RNA-Seq analysis in P. monodon stomach showed up-regulation of three THBSs during AHPND [12]. Our RNA-seq data show up-regulation of thrombospondin, and protein-protein networks show THBS as part of the top-20 of the hub genes in the *network 3*. Furthermore, qPCR gene expression analysis shows an increased expression of *LvTHBS* at 3 and 48 hpi suggesting that *LvTHBS* may be involved in the defense mechanism during AHPND.

One of the most frequent proteins up-regulated during bacterial infections is the cellular apoptosis susceptibility protein (CAS), CAS is a nuclear transport factor associated tumor necrosis factor (TNF)-mediated apoptosis, studies have suggested that in healthy cells, CAS acts as a switch of proliferation or apoptosis [76], and it has been linked to invasion and metastasis of cancer cells [77]. In other animals, *CAS* is up-regulated in non-specific cytotoxic cells in response to apoptosis [78]. Crustacean *CAS* in *F. chinenesis* has been related to ovary development and suggested to function as a nuclear protein. Moreover, CAS upregulation has been shown in *Erichor sinensis* hemocytes after *Spiroplasma eriocheris* challenge [79,80]. Furthermore, hemocytes response during AHPND has shown activation of the apoptosis gene *CAS2* in *L. vannamei* [81]. Our RNA-seq and gene expression analysis shows up-regulation of *LvCAS*, suggesting that the apoptosis process is an essential mechanism response to AHPND.

In addition to the immune-related cluster, we also looked for metabolism-related DETs in our RNA-Seq data. A connection between the immune system and metabolism (immunometabolism) has not been demonstrated in vertebrates yet; this is mainly due to the highly specialized organ structures and their complex functions in both systems. However, invertebrates possess hepatopancreas, the evolutionary forerunner of liver and pancreas, a much more simpler organ that integrates both, metabolic and immune functions, which is an ideal model to study immunometabolism [82].

One of the main enzymes involved in gluconeogenesis is the phosphoenolpyruvate carboxykinase (PEPCK), in hypoxic conditions, PEPCK expression increases to keep glucose homeostasis within the cells [83]. *Vibrio* infections can decrease oxygen uptake during pathogenesis

and induce hypoxia [84]. Our RNA-seq results and gene expression analysis show up-regulation of PEPCK expression by 17-fold at 24 hpi. The *LvPEPCK* overexpression is probably related to 1) the hypoxia conditions and lack of oxygen uptake due to the *Vibrio* infection and 2) the difficulty in maintaining glucose levels, during AHPND shrimp cannot feed and use other intermediaries metabolites to biosynthesize *de Novo* glucose [85].

Cellular oxygen deprivation alters metabolic pathways resulting in a hypoxia response, to prevent further damage due to low oxygen levels, cells protect themselves with a group of chaperone proteins called oxygen-regulated proteins [86]. Interestingly, the hypoxia gene *LvHYO* was up-regulated at 6 and 24 h in the infected group (Fig 9C). Over-expression of hypoxia genes in bacterial pathogenesis has been related to an increase in bacterial load during early AHPND stages causing hepatopancreatic septicemia, probably due to the high levels of PirAB toxins in the cells [5,11].

Chymotrypsin (CTR) is a serine-protease enzyme that participates in immune reactions and other physiological functions [87]. The synthesis of CTR is located in hepatopancreas tubules, and mRNA levels of *CTR* increase after infection with *V. anguillarum* in *F. chinensis*. Interestingly, our RNA-seq data and qPCR analyzes also show increased *LvCTR* transcript levels at 24 hpi suggesting that *LvCTR* expression is important not only to maintain the normal cellular functions but also in the immune response against *Vibrio* infections, although additional tests will be needed to determine the exact function of CTR during AHPND in shrimp [88].

Cathepsins (CTSHs) are proteases that maintain cellular turnover, acting as scavenger proteins and food digestive. Cathepsin L in shrimp is associated with rapid cell differentiation of F and B-cells in hepatopancreas [89,90]. Our data show down-regulation of *LvCTSH* at 24 hpi; however, there were no significant differences in expression in any of the time point tested, suggesting that *LvCTSH* might not play an important role during AHPND. However, as mention above, we cannot rule out the importance of *LvCTSH* expression in keeping the cell differentiation in the tubule structure especially since our histopathological analysis shows epithelial cells sloughing at 12 hpi.

As mentioned, several studies have looked at the pathogen- host response in shrimp using either different shrimp species or pathogens [5,12,15,75,81]. However, very few studies have investigated the shrimp immunological response at the transcriptome level. For example, Ge et al. (2017) which used the ridgetail white prawn (*E. carinicauda*) and *V. parahaemolyticus* to challenge the specimens by intramuscular injection found up- regulation of the hypoxia up-regulated gene (*HCY*) and chymotrypsin (*CTR*), and down- regulation of the Cathepsin L protein (*CTSH*) gene [15], these data agree with our results, we also found that these three genes have the same gene expression dynamics after bacteria challenge. Furthermore, we both found very similar enriched GO terms including, antimicrobial peptides, lectins, serine-protease cascade, oxidative-stress, pattern recognition proteins, coagulation, and heat-shock proteins (S2 and S3 Tables). A second study performed by Soonthornchai and colleagues (2016) showed that peroxiredoxin (*PRDX*), the cottable protein (*CLO*) gene and thrombospondin (*THBS*) were up- regulated in the black tiger shrimp (*P. monodon*) stomach after *V. parahaemolyticus* infection, which are exactly the same up-regulated genes found in our *L. vannamei* hepatopancreas samples after infection [12].

In addition to identifying genes involved in the host immune response, like in both previous studies, we also looked for genes that participate in metabolic pathways. Bacterial infections tend to modify their host metabolism via homeostasis disruption [91]. Based upon our analysis, we identified a new set of differentially expressed metabolic genes, among them: the antioxidant superoxide dismutase (*SOD*), hemocyanin (*HCY*), the cellular apoptosis susceptibility protein (*CAS*) gene, phosphoenolpyruvate carboxykinase (*PEPCK*), chymotrypsin (*CTR*), and the hypoxia up-regulated (*HYO*) gene. Interestingly, none of these genes have

previously been reported as being differentially expressed during pathogen infection. Determining the function of these genes during the metabolic response will provide us with a deeper understanding of how the metabolic and immune system interact and work towards specific pathogens.

## Conclusions

In conclusion, we have constructed a *de Novo* transcriptome of *L. vannamei* hepatopancreas infected with *V. parahaemolyticus* (AHPND) at 24 h using the Hi-Seq 2500 Illumina platform. We found a total of 915 differentially expressed transcripts, from which 442 were up and 473 down-regulated, up- and down-regulated transcripts were found enriched in different categories: metabolic process, oxidation-reduction process, regulation of programmed cell death, biological adhesion mainly located in the extracellular matrix. Protein-protein network analysis showed interactions between up and down-regulated transcripts within the same category, for instance in the cellular process group, response to stress genes were turned on whereas intracellular transport genes were turned off. This protein network analysis also allowed us to select transcripts with the highest number of interactions between the up-and-down-regulated clusters, a clear example is the *LvTHBS* gene, which due to its interaction and position in the network was defined as a hub gene.

Based on our enriched GO terms results and previous transcriptomic published data, we selected ten candidate genes associated to digestion, metabolism and the immune system to perform qPCR. Gene expression results of the candidate genes showed that each gene has a specific expression dynamics at the different times points analyzed, which agrees with our RNA-seq differential expression data. It will be necessary to perform proteomics analysis to study in depth how the metabolic and immunologic regulation works in shrimp during the AHPND since this knowledge can be the basis for designing therapeutic strategies. Our GO data, protein-protein networks and gene expression analyses strongly suggest a close interaction between the hepatopancreas metabolism and immune system, indicating that both systems respond collectively against a *Vibrio parahaemolyticus* infection.

## Supporting information

**S1 Fig. Agarose gel with PCR products of genes PHP and PirB.** 2 and 5 lane shows *Vibrio parahaemolyticus* IPNGS16, 3 and 6 lane shows *Vibrio parahaemolyticus* CAIM170 (Non-AHPND). 4 and 7 lane shows negative control (water). Lane 1 shows molecular marker 1 kb plus.
(TIF)

**S2 Fig. Interaction networks of DETs predicted by GeneMANIA.** The Networks represent protein- protein interactions. A) *Network 1* for up-regulated transcripts. B) *Network 2* for down-regulated transcripts and C) *Network 3* a combination of up and down-regulated transcripts. Red triangles represent up-regulated transcripts, blue inverted chevrons represent down-regulated transcripts, and gray circles represent GeneMANIA predicted proteins for each network.
(TIF)

**S3 Fig. RNA-seq data validation.** Time 0 h post-infection vs. 24 h post-infection.
(TIF)

**S1 Table. MixS descriptors.**
(XLSX)

**S2 Table. List and annotation of up-regulated transcripts.**
(XLSX)

**S3 Table. List and annotation of down-regulated transcripts.**
(XLSX)

**S4 Table. Up-regulated biological process enriched terms of *Network 1*.**
(XLSX)

**S5 Table. Down-regulated biological process enriched terms of *Network 2*.**
(XLSX)

**S6 Table. Up- and down-regulated biological process enriched terms of *Network 3*.**
(XLSX)

## Acknowledgments

This research was supported by grants of Consejo Nacional de Ciencia y Tecnología (CONA-CyT) Problemas Nacionales 247567 to NE and from CIBNOR (PAC) To FA. AEVL have a scholarship from CONACyT No. 466639. Mayra de Fátima Vargas Mendieta and Jesús Antonio Aguilar Villavicencio for technical support in organism maintenance. Eulalia Meza Chávez of Histology Laboratory in CIBNOR. Dr. Ricardo Vázquez-Juárez of bioinformatics laboratory. Rosa Maria Morelos-Castro for technical support for gene expression analysis.

## Author Contributions

**Conceptualization:** Adrián E. Velázquez-Lizárraga, Ilie S. Racotta, Humberto Villarreal-Colmenares, Oswaldo Valdes-Lopez, Norma Estrada, Felipe Ascencio.

**Data curation:** Adrián E. Velázquez-Lizárraga, Oswaldo Valdes-Lopez, Felipe Ascencio.

**Formal analysis:** Adrián E. Velázquez-Lizárraga, Norma Estrada, Felipe Ascencio.

**Funding acquisition:** Norma Estrada, Felipe Ascencio.

**Investigation:** Carmen Rodríguez-Jaramillo, Felipe Ascencio.

**Methodology:** Adrián E. Velázquez-Lizárraga, José Luis Juárez-Morales, Ilie S. Racotta, Oswaldo Valdes-Lopez, Antonio Luna-González, Carmen Rodríguez-Jaramillo, Norma Estrada, Felipe Ascencio.

**Project administration:** Norma Estrada, Felipe Ascencio.

**Resources:** Adrián E. Velázquez-Lizárraga, Antonio Luna-González, Norma Estrada, Felipe Ascencio.

**Software:** Adrián E. Velázquez-Lizárraga, Carmen Rodríguez-Jaramillo.

**Supervision:** Ilie S. Racotta, Humberto Villarreal-Colmenares, Antonio Luna-González, Carmen Rodríguez-Jaramillo, Norma Estrada, Felipe Ascencio.

**Validation:** Adrián E. Velázquez-Lizárraga, José Luis Juárez-Morales, Humberto Villarreal-Colmenares, Oswaldo Valdes-Lopez, Norma Estrada, Felipe Ascencio.

**Visualization:** Adrián E. Velázquez-Lizárraga, José Luis Juárez-Morales, Ilie S. Racotta, Oswaldo Valdes-Lopez, Carmen Rodríguez-Jaramillo, Norma Estrada, Felipe Ascencio.

**Writing – original draft:** Adrián E. Velázquez-Lizárraga, José Luis Juárez-Morales.

**Writing – review & editing:** Adrián E. Velázquez-Lizárraga, José Luis Juárez-Morales, Ilie S. Racotta, Humberto Villarreal-Colmenares, Oswaldo Valdes-Lopez, Antonio Luna-González, Carmen Rodríguez-Jaramillo, Norma Estrada, Felipe Ascencio.

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
