## [Decision Letter · Decision Letter 0]

26 Jun 2019

PONE-D-19-15646

Transcriptomic analysis of Pacific White Shrimp (Litopenaeus vannamei, Boone 1931) in response to acute hepatopancreatic necrosis disease caused by Vibrio parahaemolyticus

PLOS ONE

Dear Dr. Ascencio,

Thank you for submitting your manuscript to PLOS ONE. After careful consideration, we feel that it has merit but does not fully meet PLOS ONE’s publication criteria as it currently stands. Therefore, we invite you to submit a revised version of the manuscript that addresses the points raised by the reviewers during the review process.

We would appreciate receiving your revised manuscript by Aug 10 2019 11:59PM. To enhance the reproducibility of your results, we recommend that if applicable you deposit your laboratory protocols in protocols.io, where a protocol can be assigned its own identifier (DOI) such that it can be cited independently in the future. For instructions see: http://journals.plos.org/plosone/s/submission-guidelines#loc-laboratory-protocols

We look forward to receiving your revised manuscript.

Kind regards,

Irene Söderhäll

Academic Editor

PLOS ONE

Journal Requirements:

1.

Reviewers' comments:

Reviewer's Responses to Questions

**Comments to the Author**

1. Is the manuscript technically sound, and do the data support the conclusions?

Reviewer #1: Yes

Reviewer #2: Yes

2. Has the statistical analysis been performed appropriately and rigorously? 

Reviewer #1: Yes

Reviewer #2: Yes

3. Have the authors made all data underlying the findings in their manuscript fully available?

Reviewer #1: Yes

Reviewer #2: Yes

4. Is the manuscript presented in an intelligible fashion and written in standard English?

Reviewer #1: Yes

Reviewer #2: Yes

5. Review Comments to the Author

Reviewer #1: The manuscript reports the transcriptome analysis of the L. vannamei hepatopancreas during AHPND infection. A total of 915 differentially expressed transcript were found and categorized. Protein-protein network analysis was performed and hub genes were identified. Ten selected candidate genes were further analyzed by qRT-PCR to validate the RNA-seq differential expression data. The analysis data suggest a close interaction between the hepatopancreas metabolism and immune system, indicating their important during AHPND infection. The manuscript contains informative data that lead to the understanding of host immune response to AHPND infection. A few comments and suggestions are listed below”

1. Figure legends should be separated from the Results Part. Most figure legends should be rewritten to provide adequate information of the experiments as they must be self-explanatory. Some information such as number of shrimps, infection method, no. of repeated experiment should be included.

2. It is suggested that the gene name should be labeled in Figs. 8 and 9 so the readers could easily follow the gene expression of each selected gene. The x-axis should be changed to time post infection (hpi) to correspond to the text.

3. Please add to discussion the DE genes found in this study vs those reported in previous studies. What are the differences and new finding?

4. Line 336, CL50 should be changed to LC50

Reviewer #2: In this manuscript, the authors reported transcriptomic mRNA sequencing of infected shrimp hepatopancreas by AHPND at 24 hours post-infection, in which 915 transcripts were found differentially expressed after comparative transcriptomic analysis: 442 up-regulated and 473 down-regulated transcripts. Gene Ontology term enrichment analysis for up-regulated transcripts includes metabolic process, regulation of programmed cell death, carbohydrate metabolic process, and biological adhesion, whereas for down-regulated transcripts include, microtubule-based process, cell activation, and chitin metabolic process. The analysis of protein- protein network between up and down-regulated genes (Network 3) indicates that the first gene interactions are connected to oxidation-processes and sarcomere organization. Additionally, protein-protein networks analysis identified 20- top highly connected hub nodes. Based on their immunological or metabolic function, ten candidate transcripts were selected to measure their mRNA relative expression levels in AHPND infected shrimp hepatopancreas by RT-qPCR. The authors suggested that a close connection between the immune and metabolism systems during AHPND infection. The RNA-Seq and RT-qPCR data provide the possible immunological and physiological scenario as well as the molecular pathways that take place in the shrimp hepatopancreas in response to an infectious disease. As the AHPND is currently caused a big disease problem for shrimp culture but the molecular interaction between host and bacteria is limited. Hence, this study provides meaningful data for the research of shrimp-AHPND interactions, and probably also useful target for anti-AHPND control.

6. PLOS authors have the option to publish the peer review history of their article (what does this mean?). If published, this will include your full peer review and any attached files.

Reviewer #1: No

Reviewer #2: Yes: Hai-peng Liu

---

## [Author Response · Author response to Decision Letter 0]

24 Jul 2019

Dear Reviewers

We greatly appreciate the revision of our manuscript and thank the reviewers for their generous comments on the manuscript. We have edited the manuscript to address their concerns. In particular the comments from reviewer #1 and comments related to PLOS ONE style. The response to the reviewers' concerns is detailed on the document named as 'Response to Reviewers'.

We believe that the manuscript is now suitable for publication in PLOS ONE journal.

---

## [Editor Report · Decision Letter 1]

29 Jul 2019

Transcriptomic analysis of Pacific White Shrimp (Litopenaeus vannamei, Boone 1931) in response to acute hepatopancreatic necrosis disease caused by Vibrio parahaemolyticus

PONE-D-19-15646R1

Dear Dr. Ascencio,

We are pleased to inform you that your manuscript has been judged scientifically suitable for publication and will be formally accepted for publication once it complies with all outstanding technical requirements.

With kind regards,

Irene Söderhäll

Academic Editor

PLOS ONE
---

## [Editor Report · Acceptance letter]

2 Aug 2019

PONE-D-19-15646R1 

Transcriptomic analysis of Pacific White Shrimp (*Litopenaeus vannamei*, Boone 1931) in response to acute hepatopancreatic necrosis disease caused by *Vibrio parahaemolyticus*

Dear Dr. Ascencio:

I am pleased to inform you that your manuscript has been deemed suitable for publication in PLOS ONE. Congratulations! Your manuscript is now with our production department. 

With kind regards,

on behalf of

Dr. Irene Söderhäll 

Academic Editor

PLOS ONE